# Gait and Balance Assessments with Augmented Reality Glasses in People with Parkinson’s Disease: Concurrent Validity and Test–Retest Reliability

**DOI:** 10.3390/s24175485

**Published:** 2024-08-24

**Authors:** Jara S. van Bergem, Pieter F. van Doorn, Eva M. Hoogendoorn, Daphne J. Geerse, Melvyn Roerdink

**Affiliations:** Department of Human Movement Sciences, Faculty of Behavioural and Movement Sciences, Vrije Universiteit Amsterdam, Amsterdam Movement Sciences, 1081 BT Amsterdam, The Netherlandsm.roerdink@vu.nl (M.R.)

**Keywords:** augmented reality, Parkinson’s disease, gait and balance tests, concurrent validity, test–retest reliability

## Abstract

State-of-the-art augmented reality (AR) glasses record their 3D pose in space, enabling measurements and analyses of clinical gait and balance tests. This study’s objective was to evaluate concurrent validity and test–retest reliability for common clinical gait and balance tests in people with Parkinson’s disease: Five Times Sit To Stand (FTSTS) and Timed Up and Go (TUG) tests. Position and orientation data were collected in 22 participants with Parkinson’s disease using HoloLens 2 and Magic Leap 2 AR glasses, from which test completion durations and durations of distinct sub-parts (e.g., sit to stand, turning) were derived and compared to reference systems and over test repetitions. Regarding concurrent validity, for both tests, an excellent between-systems agreement was found for position and orientation time series (ICC_(C,1)_ > 0.933) and test completion durations (ICC_(A,1)_ > 0.984). Between-systems agreement for FTSTS (sub-)durations were all excellent (ICC_(A,1)_ > 0.921). TUG turning sub-durations were excellent (turn 1, ICC_(A,1)_ = 0.913) and moderate (turn 2, ICC_(A,1)_ = 0.589). Regarding test–retest reliability, the within-system test–retest variation in test completion times and sub-durations was always much greater than the between-systems variation, implying that (sub-)durations may be derived interchangeably from AR and reference system data. In conclusion, AR data are of sufficient quality to evaluate gait and balance aspects in people with Parkinson’s disease, with valid quantification of test completion durations and sub-durations of distinct FTSTS and TUG sub-parts.

## 1. Introduction

Impaired posture, gait, and balance control are motor symptoms commonly observed in people with Parkinson’s disease [1]. Assessing such motor symptoms may offer healthcare professionals valuable insight into disease progression and patients’ daily life challenges, which may inform tailored treatment strategies. To this end, the Five Times Sit To Stand (FTSTS) and Timed Up and Go (TUG) tests are commonly performed [2,3]. In FTSTS, the test person is asked to stand up and sit down from a chair five times as quickly as possible with their arms crossed. Longer FTSTS completion times are associated with reduced leg muscle strength, impaired balance, and increased fall risk [4]. TUG is probably the most used clinical gait and balance test, combining distinct gait and balance aspects in a single test like transfers, gait initiation, walking, and turning. In TUG, the test person is asked to stand up from an armchair, walk 3 m, turn around, walk 3 m back, and sit back in the chair. Longer TUG completion times are associated with poorer muscle strength, poorer balance, slower gait speed, and increased fall risk [5].

The standard outcome of both FTSTS and TUG is the duration of test completion, as measured by the clinician with a stopwatch. The drawback of this method is that handling the stopwatch may hinder the clinician from fully concentrating on observing the patient for safety and visual assessment. Moreover, the stopwatch score only provides an indication of overall test completion durations while remaining blind for specific limitations in distinct sub-parts of the test, like turning or sit to stand parts. To alleviate such inherent drawbacks, (automated) instrumented tests have been introduced using sensor technology, such as body-worn IMU sensors and external marker-based or markerless 3D motion registration systems (e.g., [6,7]), providing information about the execution of (sub-parts of) the tests that may allow for a more specific assessment and treatment of motor impairments. 

Augmented reality (AR) glasses, like Microsoft’s HoloLens 2 and Magic Leap 2, represent a promising emerging technology for two reasons. First, they act as a movement registration system, uniquely providing both position and orientation data in 3D using visual Simultaneous Localization and Mapping (vSLAM) algorithms [8], and second as an instrument to potentially (self-)administer tests in a standardized manner, using 3D holographic AR content to set test constraints (e.g., present a holographic pylon at 3 m from the chair indicating where the test person should turn in the TUG test) and to provide standardized instructions (e.g., ‘stand up from the chair and complete the test in 3-2-1-go’). These capabilities can enhance the evaluation of gait and balance by providing more precise and detailed measurements of the movements during (sub-parts of) the tests, thereby potentially identifying specific changes in motor function that could be critical for specific intervention strategies. Early research with healthy adults by Sun and colleagues [9] has already explored the potential of 3D position and orientation data of HoloLens 1 for deriving TUG test completion durations in comparison to stopwatch-based durations (with excellent between-systems agreement) and durations derived from IMU data (with good between-systems agreement). More recent work with healthy adults by Koop and colleagues [10] demonstrated statistical equivalence between TUG turning parameters derived from HoloLens 2 AR data and 3D motion registration data as a reference. Despite their apparent potential, AR glasses have not been validated for assessing TUG and FTSTS in clinical populations, like people with Parkinson’s disease. 

The objective of this study was to evaluate concurrent validity and test–retest reliability of FTSTS and TUG tests in people with Parkinson’s disease using HoloLens 2 and Magic Leap 2 AR glasses. Specifically, we first examined the agreement between AR position and orientation time series and counterparts from reference motion registration systems, for which a good-to-excellent agreement is expected. Subsequently, we derived test completion durations as well as sub-durations for distinct sub-parts of the tests from these time series and evaluated concurrent validity in terms of between-systems absolute-agreement statistics (i.e., intraclass correlation coefficient (ICC), bias, limits of agreement) of test completion durations derived from AR data and the stopwatch (as clinical gold standard), as well as the between-systems absolute agreement of AR (sub-)durations with counterparts from reference systems. Finally, to help interpret the so-obtained between-systems absolute agreement statistics, we determined within-system test–retest reliability in terms of ICC, bias, and limits of agreement. We expected that (sub-)durations can be validly and reliably derived from AR data in people with Parkinson’s disease, with better between-systems than within-system absolute agreement statistics.

## 2. Materials and Methods

### 2.1. Subjects

A convenience sample of 22 subjects diagnosed with Parkinson’s disease, who were capable of walking independently for over 30 min, participated in this study. See Table 1 for a detailed overview of participant characteristics. Participants did not have any other neurological or orthopedic conditions that would significantly affect their walking ability. Their cognitive function was sufficient to understand the instructions provided by the researchers. Participants did not report experiencing hallucinations and had no visual or hearing impairments. 

### 2.2. Experimental Set-Up and Procedures

Participants maintained their regular daily medication schedule and were invited for one visit to the gait laboratory at the Vrije Universiteit Amsterdam (VU). Participants completed the FTSTS test first, followed by the TUG test during one measurement session, while wearing either HoloLens 2 (*N* = 12) or Magic Leap 2 (*N* = 10) AR glasses, block randomized over participants. To evaluate test–retest reliability of AR data-derived (sub-)durations, both the FTSTS and TUG tests were performed for a second time in the same measurement session, maintaining the order of first the FTSTS test and then the TUG test. Participants initiated the FTSTS and TUG tests from a seated position with their backs touching the chair and concluded the test in the same position. A stopwatch was used to register the durations of FTSTS and TUG test completion. For FTSTS, participants were instructed to stand up and sit down five times in a sequential manner as quickly as possible, while keeping their arms crossed on their chest, without touching the backrest of the chair for all but the last sit down movement (Figure 1a). For TUG, participants were asked to perform the following sequence of actions: stand up from a standard armchair, walk 3 m, turn around, walk back to the chair, and finally, transfer from a standing to a sitting position with a turn (Figure 1b). Additionally, two reference motion registration systems were used to evaluate the concurrent validity of the AR time series and AR data-derived (sub-)duration outcomes: (1) a Microsoft Kinect v2 sensor (Kinect) as part of the Interactive Walkway (Tec4Science, VU Amsterdam, [12,13]) to record 3D position data of various body points, of which we used the head, sternum, and spine base and (2) an Inertial Measurement Unit (IMU; McRoberts B.V., The Hague, The Netherlands) worn on the lower back to record trunk orientation. None of the participants experienced freezing of gait during the tests, as observed by the researchers.

### 2.3. Data Acquisition

HoloLens 2 and Magic Leap 2 are state-of-the-art AR glasses registering their 3D position and orientation with regard to their surroundings at a sampling rate of 30 and 60 Hz, respectively. Specific 3D position and orientation time series contain features that are informative for distinguishing various sub-parts of the tests, like standing up, sitting down, and turning (as detailed in Section 2.4). In the Appendix A, videos are provided of the TUG and FTSTS tests’ performance, including a synchronized visualization of pertinent AR data. As a reference, we acquired data from IMU and Kinect sensor systems. Specifically, the IMU captured at 100 Hz the 3D accelerations and trunk rotation velocity time series, from which reference turning sub-durations of TUG were derived (as detailed in Section 2.4). The Microsoft Kinect computer vision sensor captured at 30 Hz in a markerless manner the 3D positions of various bodily points, including the head, sternum, and spine base, from which reference (sub-)durations of the FTSTS test were derived (as detailed in Section 2.4). 

### 2.4. Data (Pre)Processing

The time series from each of the three motion registration systems was resampled to a constant rate of 60 Hz using linear interpolation and low-pass filtered using a fourth-order Butterworth filter with a cut-off frequency of 2 Hz. Temporal alignment in between-systems time series was obtained by incorporating the time lag of their maximal cross-correlation. The initial starting positions and orientations were subtracted from the time series. 

#### 2.4.1. Deriving (Sub-)Durations of the FTSTS Test

The AR and Kinect vertical position time series were used to determine the start and end of the FTSTS test (standing up and sitting down in a chair; Figure 2, vertical dashed lines). Specifically, we calculated zero crossings in the vertical velocity time series to obtain an initial indication of the start and end of the test. It is important to realize that standing up and sitting down are not strict upward and downward movements, as the initial standing up movement typically involves a forward bow [1,14] while the final sitting down movement involves a backward bow associated with placing the back against the backrest of the chair. Hence, to obtain a more representative start and end of the FTSTS test, we determined characteristic transition points in the vertical position data (instead of simply their minima) using a mathematical model consisting of the following piecewise linear function to find the definite start (Equation (1)): (1)$$y\left(x;\;b\right)\;=\;b_{1}+b_{2}x\;+\;b_{3}\left(x-b_{4}\right)H\left(x-b_{4}\right)$$
where *H*(*x* − *b*_4_) is the step function defined as follows:(2)$$H\left(x-b_{4}\right)\;=\;{├}_{1\;if\;x\;>\;b_{4}}^{0\;if\;x\leq \;b_{4}}$$

In this model, *b*_1_ represents the initial offset of the vertical position, *b*_2_ represents the first slope (set to 0 for identifying the start frame), *b*_3_ represents the second positive slope, and *b*_4_ represents the breakpoint. To obtain the start of the test (frame number corresponding to *b*_4_), we fitted the function (*y*) on the vertical position time series (*x*) from the first frame of the recording to the frame corresponding to the initial indication of the start of the test using a subspace trust region-based, nonlinear, least squares optimization method. A similar method was used to find the end of the test, where the breakpoint was found by fitting a piecewise linear function, starting with a positive slope and ending with a constant, to the vertical position time series from the initial indication of the end of the test to the end of the recording. With these frame numbers, the time series were trimmed from start to end, from which we determined (i) the between-systems agreement in the time series and (ii) the test completion durations for further statistical analyses.

From the trimmed data, each sit to stand cycle of FTSTS was divided into four phases: sitting, sit to stand, standing, and stand to sit. Sitting and standing phases were first identified from the vertical position data from each system by means of finding zero crossings in the vertical velocity. Since a zero crossing gives a single time point and not the duration of the sitting and standing phases, we applied an empirically found threshold of 2 cm to obtain the start and end frame numbers of standing and sitting phases [7], from which sitting, sit to stand, standing, and stand to sit sub-durations were derived (Figure 2). To reduce the influence of potential outliers in within-test repetitions of the so-obtained sub-durations (e.g., due to, for example, a participant taking a brief sitting rest during the test), the median of the 4 (for sitting sub-durations) or 5 (for standing, sit to stand, stand to sit sub-durations) within-test sub-durations was determined prior to further statistical analyses.

#### 2.4.2. Deriving (Sub-)Durations of the TUG Test

The TUG test completion duration was determined from the AR vertical position time series, as described for the FTSTS test. Sub-durations for the two turns (turn 1 and turn 2) were derived from the AR orientation time series, specifically the yaw angle (Figure 3). For the IMU, yaw angle time series were also used, obtained after integrating the associated rotational velocity time series. Determining the start and end of a turn during TUG is not straightforward, as it is characterized by a rather slow transition from one state (~0° yaw) to another (~180° yaw). We modeled this transition with the following sigmoid function *y* (Equation (3)): (3)$$y\left(t;p\right)\;=\;p_{1}\tanh\left(\frac{t-p_{2}}{p_{3}}\right)+p_{1}+p_{4}$$
where tanh is the hyperbolic tangent function defined as follows:(4)$$\tanh\left(z\right)\;=\;\frac{e^{z}-e^{-z}}{e^{z}+e^{-z}}$$

In this model, *p*_1_ represents the amplitude of the transition (usually ~180°), *p*_2_ represents the timing (or center) of the transition, *p*_3_ represents a scaling factor that adjusts the duration of the transition, and *p*_4_ represents the (potential) starting offset (in °). The sigmoid function *y* was fitted to the yaw angle time series *t* using a subspace trust region-based, nonlinear, and least squares optimization method. The fitted model *y* was then used to determine the start and end of turning (Figure 3) by applying an empirically found threshold of 10° from the start of the transition to the end for each turn as determined by *p*_4_ and *p*_4_ + *p*_1_, respectively. Turning sub-durations were derived from these indicators of the start and end of turns and used for further statistical analyses.

Concurrent validity (i.e., consistency agreement between AR and reference system time series and absolute between-systems agreement for all (sub-)durations of the FTSTS and TUG tests) and test–retest reliability (i.e., within-system absolute agreement between repetitions for all (sub-)durations of the FTSTS and TUG tests) were evaluated with the ICC_(C,1)_ and ICC_(A,1)_ [15]. ICC values greater than 0.50, 0.75, and 0.90 represent, respectively, moderate, good, and excellent agreement between systems or over repetitions [16]. As ICC values alone may give misleading impressions of agreement when there is a large between-subject variation, we complemented them with two Bland–Altman analysis statistics: (1) bias, indicating a systematic difference between systems or over repetitions, and (2) limits of agreement, indicating the precision of differences between systems or over repetitions [17]. Statistical differences between systems (i.e., between AR glasses and reference systems) and trials were evaluated using paired samples t-tests for normally distributed data, while differences between HoloLens 2 and Magic Leap 2 AR glasses were assessed using independent t-tests under the assumption of normality. Normality was checked using the Kolmogorov–Smirnov test. In cases where the data were not normally distributed, the non-parametric Wilcoxon signed-rank test was used for paired comparisons, and the Mann–Whitney U test was used for independent comparisons. All processed data used for the statistical analyses are provided in the Appendix A, from which missing values become apparent. Missing values mostly resulted from inaccurate Kinect data (i.e., storing 3D kinematics of the guarding researcher instead of the participant). Other reasons for missing values were as follows: (1) one participant did not perform the FTSTS test, (2) one participant was excluded from the between-systems comparisons because of an error in IMU recording, and (3) two participants were removed from test–retest analyses because of incomplete data for one of the two repetitions. For concurrent validity analyses, the second trial was generally used and, when this was not possible (e.g., due to the abovementioned issues with Kinect recordings), occasionally, the first trial was used to keep as many participants in the analyses. 

## 3. Results

### 3.1. Concurrent Validity

#### 3.1.1. Agreement in Time Series between AR and Reference Systems

Figure 4a shows representative examples of the AR vertical position data versus the three vertical Kinect body points of FTSTS. At a group level, the consistency agreement between AR and Kinect time series for FTSTS was excellent (ICC_(C,1)_ [95% CI] for AR vs. Kinect head: 0.992 [0.987–0.996], sternum: 0.976 [0.962–0.989], and spine base: 0.933 [0.916–0.951]). There were no significant differences between the consistency agreement scores of HoloLens 2 (0.988) and Magic Leap 2 (0.995) against Kinect head time series (*t*(15) = −1.81, *p* = 0.091), nor between HoloLens 2 (0.969) and Magic Leap 2 (0.980) against the Kinect sternum time series (*t*(15) = −0.78, *p* = 0.449) or between HoloLens 2 (0.921) and Magic Leap 2 (0.945) against the Kinect spine base reference time series (*t*(15) = −1.39, *p* = 0.185). Figure 4b shows a representative example of AR and IMU orientation data (yaw angle) of TUG. At a group level, the consistency agreement between AR and IMU time series for TUG was again excellent (ICC_(C,1)_ [95% CI], AR vs. IMU trunk: 0.986 [0.981–0.990]). Against the IMU time series, consistency agreement scores did not differ between HoloLens 2 (0.983) and Magic Leap 2 (0.989) (*t*(19) = −1.46, *p* = 0.161).

#### 3.1.2. Agreement in (Sub-)Durations Derived from AR and Reference System Data

The absolute agreement between AR-derived test completion durations and stopwatch counterparts was excellent for both the FTSTS and TUG tests (ICC_(A,1)_ > 0.984), with a small but significant bias (~2% and ~4% of the mean, respectively) and narrow limits of agreement (Table 2). 

The absolute agreement statistics between derived (sub-)durations from AR and reference system time series are presented in Table 2. For FTSTS, the between-systems agreement (AR vs. Kinect head data) for derived total and sub-durations was excellent (ICC_(A,1)_ > 0.921, with three related small but significant biases for sitting, sit to stand, and stand to sit sub-durations and narrow limits of agreement). A similar pattern of between-systems agreement statistics was observed for the comparisons of AR (sub-)durations with those derived from Kinect sternum and spine base data, with slightly worsening statistics for body points farther away from the head (see Appendix A). 

For TUG, the between-systems agreement for sub-durations was excellent for turn 1 (ICC_(A,1)_ = 0.913, without bias and narrow limits of agreement) and moderate for turn 2 (ICC_(A,1)_ = 0.589, with a substantial bias of ~22% and wide limits of agreement; Table 2).

### 3.2. Test–Retest Reliability for (Sub-)Durations of FTSTS and TUG

The absolute agreement between trial 1 and trial 2 for the AR data of the FTSTS test was excellent for total durations (ICC_(A,1)_ = 0.914) and good to excellent for sub-durations (ICC_(A,1)_ > 0.830). Only the standing sub-durations showed moderate absolute agreement between repetitions (ICC_(A,1)_ = 0.695; Table 3). All biases were small and non-significant, while limits of agreement between test and retest trials were all greater than those seen between systems (i.e., compared to Table 2). The test–retest statistics for the reference systems data were similar or slightly worse (see Appendix A).

The absolute agreement for TUG test completion durations between trial 1 and trial 2 for the AR data was good (ICC_(A,1)_ > 0.75, no bias, wider limits of agreement than those seen between systems) and moderate for turn 1 and turn 2 sub-durations (ICC_(A,1)_ < 0.75, no bias and again wider limits of agreement than those seen between systems; cf. Table 3 vs. Table 2). The test–retest statistics for the reference system data were again similar (see Appendix A). Note that we used the standard TUG test without dictating turning directions (i.e., clockwise or counterclockwise). Nevertheless, the turning directions of the test and retest were highly consistent among our participants. For turn 1 at or around the 3-m marker, all our participants turned in the same direction for the test and retest. For turn 2 prior to sitting down, 18 of the 21 participants turned in the same direction for test and retest; excluding the three participants with an inconsistent turn direction slightly improved test–retest reliability statistics for turn 2 sub-durations (limits of agreement: from −0.14 (−1.28 1.00) to −0.02 (−1.06 1.02) and ICC_(A,1)_ from 0.684 to 0.742). 

## 4. Discussion

The aim of this study was to evaluate concurrent validity and test–retest reliability of AR-instrumented FTSTS and TUG tests in people with Parkinson’s disease. Here, we discuss the findings associated with the three specific objectives outlined in the Introduction. The first objective was to examine the agreement between AR position and orientation time series and counterparts from reference motion registration systems. An excellent concurrent validity was observed, better than expected, as evidenced by excellent ICC values between AR and Kinect vertical position data and between AR and IMU yaw orientation data. This was true for both HoloLens 2 and Magic Leap 2 AR glasses, as our findings revealed no significant difference in consistency agreement scores between HoloLens 2 and Magic Leap 2 AR glasses, suggesting comparable accuracy in 3D position and orientation tracking. As can be appreciated from the data depicted in Figure 2, Figure 3 and Figure 4, as well as from the data visualization alongside videos of TUG and FTSTS performance in the Appendix A, AR 3D position and orientation data contain rich information from which indicators of various distinct sub-parts of the tests can be validly derived, as discussed further below. Although we focused on temporal aspects of TUG and FTSTS test performance, there are ample opportunities for an even finer-grained parameterization of identified sub-parts given that state-of-the-art AR glasses (i.e., HoloLens 2, Magic Leap 2) are, in principle, 3D position sensors, a unique asset compared to other wearable sensor systems [9]. Specifically, features in the vertical position time series, in the horizontal displacement time series, and in the orientation around the vertical axis (i.e., yaw angle time series) seem informative for demarcating the various sub-parts of the TUG test (Figure 5). As can be appreciated in Figure 5, the yaw angle time series (dashed line) clearly shows two ~180° turns, while the change in slope in the horizontal displacement time series (dotted line) may be indicative of deceleration and acceleration phases demarcating turning and walking phases. Deriving spatiotemporal gait parameters, like cadence and step lengths, from TUG walking parts seems well feasible using the characteristic oscillations in the vertical (and non-depicted mediolateral) position time series associated with midstance (peaks) and foot strikes (valleys) [13]. Gait parameter quantification was already successfully explored previously for healthy adults [10,18] and people with Parkinson’s disease [13] and is deemed worth studying further for standard clinical tests, like, for instance, the 10-m walk test and the 6-min walk test. 

The second objective was to derive, from these valid time series, TUG and FTSTS test completion durations and sub-durations for distinct sub-parts of the tests, like turning and sit to stand durations, and to evaluate their concurrent validity against reference systems. For TUG and FTSTS test completion durations, we found excellent agreement between AR data and the stopwatch, albeit with a small but significant bias (<4%) and narrow limits of agreement, indicating that the systems can be used interchangeably. Likewise, between-systems agreement statistics were excellent for all (sub-)durations for the FTSTS test, with high ICCs and narrow limits of agreement, yet with three small but significant biases (~6%) for sitting, sit to stand and stand to sit sub-durations, annulling each other subsequently given that they were, in absolute terms, similar in magnitude but opposite in direction, indicative of an interconnected between-systems bias in the identification of frame numbers for the start and end of the sitting phase, demarcating stand to sit, sitting, and sit to stand sub-durations. For the TUG test, the sub-durations of the two turns showed different agreement statistics. While the between-systems agreement for sub-durations for the first turn was excellent (see also Koop and colleagues [10]), the agreement scores for the second turn were moderate. This difference between the two turns may be explained by our observation that the second turn is not a distinct sub-part but overlaps with the sitting down part of the TUG test in a combined sitting down whilst turning movement. Also, turning participants sometimes looked for the location of the chair and, therefore, turned their heads (as captured with AR yaw data) before they actually turned around with their trunk (as captured with IMU yaw data). Finally, there is an ordered sequence during turning, which generally starts with the head and is followed by the trunk and is enclosed by the head again [19], which may all have affected between-systems differences in sub-durations to some extent. 

The third objective was to determine within-system test–retest reliability to help interpret the abovementioned between-systems absolute agreement statistics. An interesting observation from a statistical point of view was that the within-system test–retest variation in completion times and sub-durations (cf. Table 3) was always much greater than the between-systems variation (cf. Table 2). This was reflected in the limits of agreement, which are much wider for the test–retest evaluation (for AR and reference systems alike, see also Appendix A) than for the between-systems evaluation (i.e., compare limits of agreement between Table 2 and Table 3). On the one hand, this is positive, as test completion durations and sub-durations may then be derived interchangeably from AR and reference system data. On the other hand, the fairly large variation seen over repeated measurements, for both AR and reference systems (see also [20,21,22]), may limit their sensitivity for detecting longitudinal changes with disease progression, medication, or rehabilitation intervention [23].

Overall, it seems fair to conclude that AR 3D position and orientation data are valid and contain rich features from which (sub-)durations of TUG and FTSTS performance can be validly derived in people with Parkinson’s disease, with—for both AR and reference systems—a better between-systems than within-system absolute agreement. What are the future prospects of these findings? We envision a scenario where AR data are used to not only quantify test completion durations (similar to the stopwatch) but, given the rich information in the data and the excellent time series concurrent validity results presented here, also to provide a more comprehensive quantitative assessment of clinically relevant sub-durations, like the turn in TUG [6]. With AR, such quantitative tests may also be automatized, as test instructions (e.g., ‘3-2-1-go’) and test constraints (AR visual indicator for TUG turn at 3 m from the chair) can readily be provided. Automated test administration resulting in valid and reliable test scores is much needed and provides opportunities to improve care. That is, physical therapy is increasingly given remotely at home with AR [23,24], for which (self-)monitoring of treatment progress becomes key. Because such home-based AR gait and balance exergaming intervention programs will typically be remotely prescribed by the therapist [23,24], in principle, also AR TUG and FTSTS assessments may be prescribed as part of the intervention program. Insight into progress can so be obtained at higher intervals than the standard pre–post-intervention assessment in clinical practice or clinical research. This is relevant as multiple longitudinal assessments may help mitigate the effect of confounding factors, like daily fluctuations, thus ensuring a more reliable assessment of change over time and thereby enhancing the quality of clinical and research findings. It may also provide unique insight into dose–response relationships of prescribed AR interventions and/or concomitant medication in a time-effective and patient-friendly manner. Ultimately, remote parameterization of progress could help reduce the number of contact moments between patient and healthcare provider and also change those consults. That is, instead of administering tests, the session may be used to discuss the patient’s test results and any further needs for intervention moving forward, which is expected to increase care efficiency and patient satisfaction. Our ongoing research will proceed along those lines. 

## 5. Conclusions

From this study, it can be concluded that AR data are valid and informative for quantifying TUG and FTSTS test performance outcomes. TUG and FTSTS test completion durations, as well as various sub-durations of distinct sub-parts of the test, can be determined interchangeably from AR data and reference system data in a cross-sectional assessment in persons with Parkinson’s disease. However, the relatively poorer within-system test–retest reliability (for AR and reference systems alike) compared to between-systems agreement should be kept in mind when performing longitudinal assessments with either system.

## Figures and Tables

**Figure 1 sensors-24-05485-f001:**
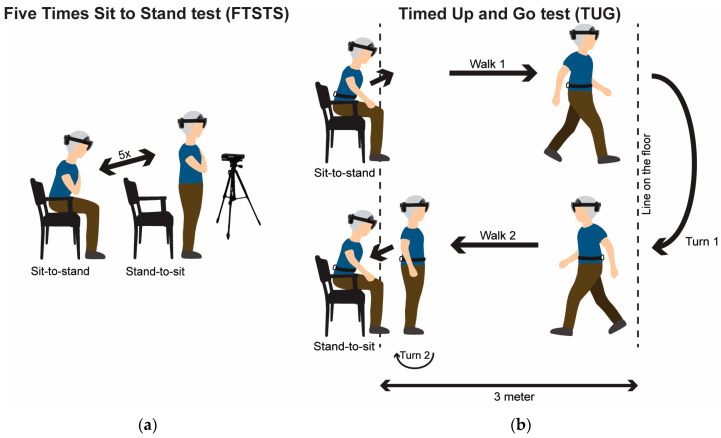
(**a**) Participant performing the Five Times Sit To Stand (FTSTS) test; data are recorded with a stopwatch, AR glasses, and a Microsoft Kinect v2 sensor. (**b**) Participant performing the Timed Up and Go (TUG) test; note that the same chair in the same position was used for standing up and sitting down; data are recorded with a stopwatch, AR glasses, and an Inertial Measurement Unit (IMU) worn on the lower back.

**Figure 2 sensors-24-05485-f002:**
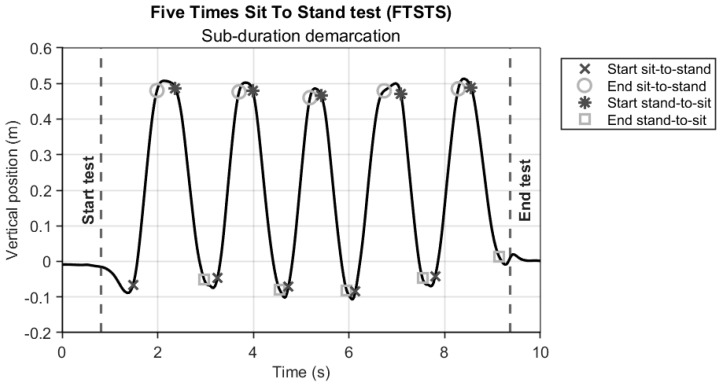
Representative AR vertical position time series of a participant performing FTSTS, with vertical dashed lines indicating the start and end of the test (from which test completion duration was derived) and with specific markers (see legend) indicating the start and end of the sit to stand, standing, stand to sit, and sitting sub-parts of the test (from which sub-durations were derived).

**Figure 3 sensors-24-05485-f003:**
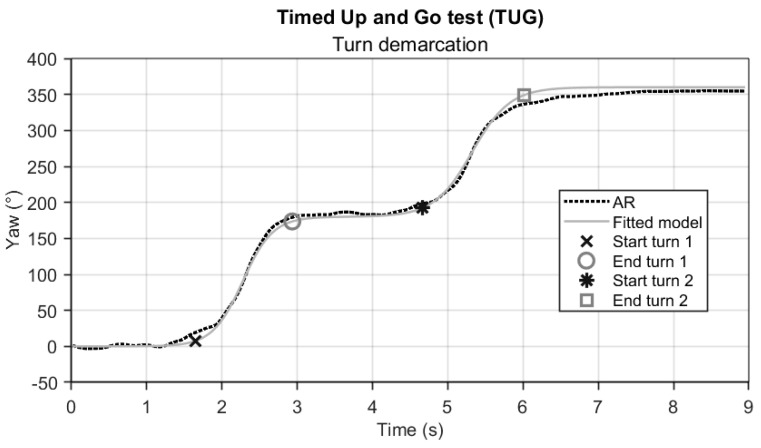
Representative AR yaw angle time series (dashed line) of a participant performing TUG, with the gray solid line representing the fitted sigmoid model *y* that was used to derive turning sub-durations. Statistical analysis.

**Figure 4 sensors-24-05485-f004:**
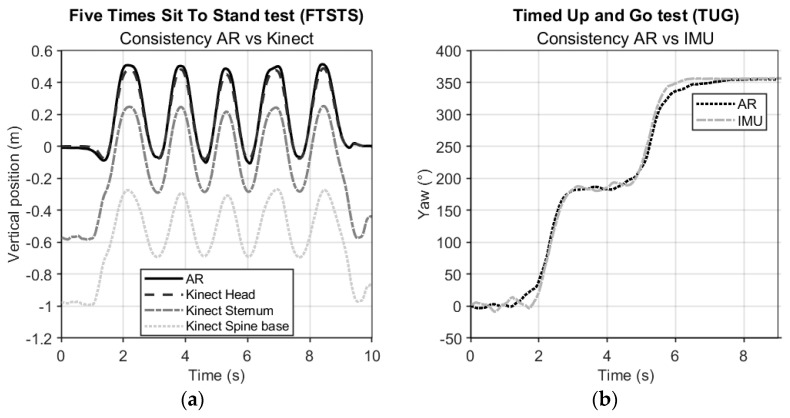
(**a**) Representative examples of AR (HoloLens 2) vertical position data vs. Kinect vertical position data during FTSTS. (**b**) Representative example of AR (Magic Leap 2) yaw orientation data vs. IMU trunk yaw data during TUG.

**Figure 5 sensors-24-05485-f005:**
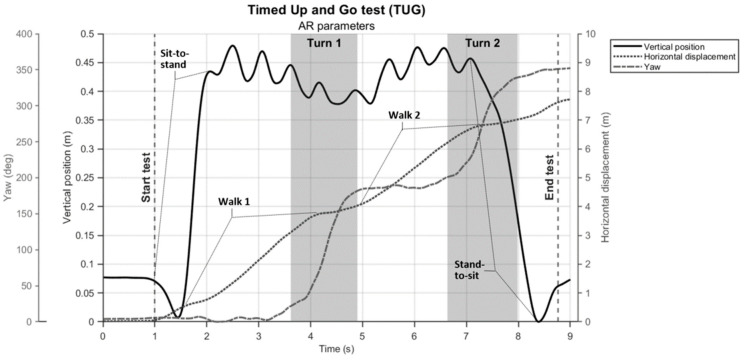
Illustration of relevant AR data time series of a participant performing the TUG test, with apparent features to identify distinct sub-parts of the test, like standing up, walking, and turning, from which various spatiotemporal parameters may be derived.

**Table 1 sensors-24-05485-t001:** Participant characteristics.

Characteristics	Data (Mean ± SD [Range] Unless Indicated Otherwise)
Age (years)	66.3 ± 8.8 [51–82]
Weight (kg)	79.1 ± 9.9 [59.0–92.9]
Height (cm)	176.1 ± 10.3 [154–191]
Sex, male/female	16/6
Diagnostic time (years)	7.5 ± 4.8 [1–20]
Modified Hoehn and Yahr stage, 2/2.5	14/8
Freezing of gait, yes/no *	11/11
MDS-UPDRS III score	31.7 ± 11.3 [13–61]
Fall history (number of falls over the previous year)	2.5 ± 3.3 [0–10]

MDS-UPDRS III = MDS-unified Parkinson’s disease rating scale part III. * The presence of freezing of gait is defined by a non-zero score on the New Freezing of Gait Questionnaire [11].

**Table 2 sensors-24-05485-t002:** Concurrent validity statistics for FTSTS and TUG (sub-)durations (in s): between-systems absolute agreement statistics for (sub-)durations derived from AR and reference systems data.

		Mean ± SD	Mean ± SD	Bias (95% Limits of Agreement)	*t*-Statistics	ICC_(A,1)_
		**AR**	**Stopwatch**			
**FTSTS**	**Completion duration**	12.3 ± 3.7	12.1 ± 3.7	−0.25 (−0.66 0.17)	*t*(20) = −5.28, ***p* < 0.001**	0.996
**TUG**	**Completion duration**	10.3 ± 3.1	9.9 ± 3.2	−0.44 (−1.19 0.31)	*t*(21) = −5.37, ***p* < 0.001**	0.984
		**AR**	**Kinect Head**			
**FTSTS ***	**Completion duration**	11.9 ± 3.91	11.91 ± 3.81	−0.04 (−0.38 0.3)	*t*(17) = −1.09, *p* = 0.293	0.999
	**Sitting sub-duration**	0.69 ± 0.49	0.62 ± 0.46	−0.07 (−0.25 0.11)	*t*(16) = −3.13, ***p* = 0.006**	0.971
	**Sit to stand sub-duration**	0.58 ± 0.16	0.62 ± 0.16	0.04 (−0.03 0.12)	*t*(16) = −4.49, ***p* < 0.001**	0.943
	**Standing sub-duration**	0.34 ± 0.15	0.32 ± 0.16	−0.02 (−0.09 0.06)	*t*(16) = 1.82, *p* = 0.087	0.965
	**Stand to sit sub-duration**	0.61 ± 0.16	0.65 ± 0.16	0.04 (−0.07 0.14)	*t*(16) = −3.05, ***p* = 0.008**	0.921
		**AR**	**IMU Trunk**			
**TUG**	**Turn 1 sub-duration**	1.75 ± 0.64	1.8 ± 0.70	−0.09 (−0.4 0.6)	*t*(20) = −1.51, *p* = 0.146	0.913
	**Turn 2 sub-duration**	2.01 ± 0.71	1.61 ± 0.62	−0.40 (−1.44 0.64)	*t*(20) = 3.44, ***p* = 0.003**	0.589

Significant biases are presented with the *p*-values **in bold**. * One outlier participant (p4 in the Appendix A) was excluded from the sub-duration results due to insufficient peaks in the Kinect time series only, preventing our peak detection algorithm from accurately identifying the frame number start and end indices.

**Table 3 sensors-24-05485-t003:** Test–retest reliability for FTSTS and TUG: absolute agreement statistics for (sub-)durations (in s) derived from AR data.

		Mean ± SD	Mean ± SD	Bias (95% Limits of Agreement)	*t*-Statistics	ICC_(A,1)_
		**AR Trial 1**	**AR Trial 2**			
**FTSTS**	**Completion duration**	12.82 ± 4.33	12.28 ± 3.83	−0.54 (−3.78 2.70)	*t*(19) = 1.45, *p* = 0.163	0.914
	**Sitting sub-duration**	0.78 ± 0.45	0.75 ± 0.48	−0.03 (−0.28 0.22)	*t*(19) = 1.04, *p* = 0.314	0.964
	**Sit to stand sub-duration**	0.61 ± 0.19	0.59 ± 0.15	−0.02 (−0.21 0.17)	*t*(19) = 1.02, *p* = 0.322	0.830
	**Standing sub-duration**	0.37 ± 0.15	0.38 ± 0.18	0.01 (−0.25 0.26)	*t*(19) = -0.32, *p* = 0.756	0.695
	**Stand to sit sub-duration**	0.61 ± 0.16	0.62 ± 0.15	0.01 (−0.13 0.14)	*t*(19) = -0.54, *p* = 0.598	0.903
**TUG**	**Completion duration**	10.83 ± 2.71	10.32 ± 3.21	−0.51 (−3.33 2.32)	*t*(20) = 1.62, *p* = 0.122	0.874
	**Turn 1 sub-duration**	1.82 ± 0.73	1.78 ± 0.63	−0.04 (−1.05 0.98)	*t*(20) = 0.34, *p* = 0.740	0.721
	**Turn 2 sub-duration**	2.18 ± 0.76	2.04 ± 0.71	−0.14 (−1.28 1.00)	*t*(20) = 1.07, *p* = 0.295	0.684

## Data Availability

Data supporting reported results can be found in the Appendix A.

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
