# Peer review of "Gait and Balance Assessments with Augmented Reality Glasses in People with Parkinson’s Disease: Concurrent Validity and Test–Retest Reliability"

_sensors, 2024, doi:10.3390/s24175485_

Round 1

Reviewer 1 Report

Comments and Suggestions for Authors

This study was a paper that evaluated walking and balance using AR glasses and verified their validity and reliability. Using AR glasses to evaluate physical functions is highly innovative and is an important study that will improve the ease of evaluation. However, there are some points that need to be revised, so please take them into consideration.

Abstract:

It would be good to include statistical values for validity and reliability.

Keywords:

The character "a" is in bold.

Introduction:

P1. Line 32-33. You need to provide evidence that TUG and FTSTS are commonly used, so you need to cite your sources.

Materials and Methods

P2, line 86. Is 22 a proper sample size?

P2, line 86. As basic attributes, the PD duration, drug dose, etc. should be added. In addition, the results of the basic attributes should be added to the result section as a table.

P2, Are these measurements in the "on" or "off" phase? Be clear.

P2, line113. Were the two measurements taken consecutively or with an interval in between?

P2, line113. In addition, patients with PD also exhibit a tendency to Freezing of gait. Therefore, please add the evaluation results of whether or not freezing of gait should be included.

P3. About TUG. In Figure 1, there is no object during rotation, and it appears that the range of rotation is wide. What is the distance of the range of rotation? What is the direction of rotation? The side of the body where Parkinson's disease first appears is more likely to experience a decline in function. Therefore, the speed should differ depending on the direction of rotation. If you are considering the physical function of Parkinson's disease, shouldn't you compare data from both directions of rotation?

P3, lime 115. In addition, the TUG uses an armchair. However, it is not used in this study. Is this the TUG? Did you decide the direction for the second TUG turn? If so, it is a different method from the regular TUG.

Results

The author reports ICC, but it would be worth considering adding a 95% confidence interval.

Comments on the Quality of English Language

No comment.

Reviewer 2 Report

Comments and Suggestions for Authors

Thank you for the opportunity to review this paper. 

The authors provide an excellent overview of the clinical tasks and relevance to clinical practice alongside the implications of poor performance on these tasks. They point to the shortcomings of the current data acquisition method (overall time with a stopwatch) which points to the need for improved data collection methods that can segment the tasks into phases. This is especially useful for the TUG task, in particular in people with PD, where freezing may occur in one phase of the task (e.g., turning) which would drive up the time to completion but would not tell a clinician or medical reviewer where the impairment occurred during the trial to warrant targeted intervention. This is clearly laid out by the authors and justifies the need to validate and assess reliability of these novel AR glasses in a clinical population where kinematic data is different from a healthy population where validation data currently exists in this space. The data pre-processing and deriving of sub phases are documented well and clearly defined for others to replicate this work. Again, the authors document a need for improved clinical outcome measurement techniques and the value it will add to clinical care in the discussion. 

Major Revision(s):

- In section 2.2, the authors note participants used either the Microsoft HoloLens 2 or the Magic Leap 2 for data collection. In section 2.3, it is addressed that these systems have differing sampling frequencies, and to my knowledge the Magic Leap 2 has not undergone a validation against the gold standard for biomechanical analysis, 3D motion capture. With that, the analysis should be re-run including only data from a single AR headset, ideally the HL2. Mixing both headsets into a single analysis does not adequately inform the reader if one data source (AR headset) is providing valid biomechanical data. 

- If the headset use was split 50/50 between participants, the authors could display a HL2 and Magic Leap 2 analysis. Knowing the HL2 provides valid biomechanical data this would provide an initial validation of the Magic Leap 2's capabilities. 

Minor Revision(s):

- Section 2.2 and Figure 1b display a deviation of the TUG task from the standard clinical test. The turn following the first straight line walk appears to lack an object or digital pylon in AR for the participant to perform the 180 degree turn around. This may have produced a more arcing turn which lacks the angular velocity that turn 2 has. It is appreciated that the researchers noted the change in head position when the participants were looking for the chair as well as being unable to fully parse out turning and sitting in turn 2.

-A power analysis should be provided to justify the sample size of 22 participants. Or potentially, a reduced sample size if there is an adequate sample to report data from a single AR device.

Reviewer 3 Report

Comments and Suggestions for Authors

This study investigated the validity and reliability of gait balance control evaluation with AR glasses in people with Parkinson's disease. Some revisions should be made as follows:

Abstract:

  • The author should start with more background information.
  • Although it is in the abstract section, some main findings, including statistical comparison values, should be provided.

Introduction:

  • The author mentioned the advantages of AR devices but missed connecting using AR to evaluate gait balance control and other fall risk prevention measures. More rationale should be added to strengthen the article's significance.

Materials and Methods:

  • Lines 86–87: The author only provided the mean for the subjects' general information. The standard deviation should also be provided.
  • Line 122: The author used different sampling rates for the two devices. Can you explain why two different sampling frequencies were chosen? Will this result in any outcome measurement discrepancies? If so, please mention this in the limitations section.

Results:

  • The author should consider updating all figures to color versions. The current black and grey ones make it difficult to identify some key points.

Comments on the Quality of English Language

Minor check

Round 2

Reviewer 1 Report

Comments and Suggestions for Authors

This study lacks basic information to ensure the validity of the AR system.

The most serious problem is that it does not assess the ON and OFF timing of Parkinson's disease. This is a major problem for the validity and reliability of the results, and as far as your response is concerned, publishing this study in its current state would be misleading to the results.

Secondly, the information on medication and the assessment of the extent of the scurvy is also vague; as you measure TUG, you should have indicated the extent of the scurvy, not whether it was present or not.

Finally, the TUG is not oriented. This leads to a lack of adjustment for confounding factors, which would make it difficult to state the results clearly.

Comments on the Quality of English Language

No comment.

Author Response

Comment 1: This study lacks basic information to ensure the validity of the AR system. The most serious problem is that it does not assess the ON and OFF timing of Parkinson's disease. This is a major problem for the validity and reliability of the results, and as far as your response is concerned, publishing this study in its current state would be misleading to the results.

Response: With all due respect, not assessing the ON and OFF timing of Parkinson's disease is NOT a major problem for our validity and reliability results. The study's main aim was to examine concurrent validity of AR data and therefrom derived outcomes against those of concurrently recorded reference systems. For this part of the study, it is irrelevant what the ON/OFF timing was as data is concurrently gathered from AR and reference systems. For the study's aim regarding test-retest reliability, the reviewer's concern becomes more relevant, yet in our view, it is not critically affecting the reliability of the results because the measurements were done in the same session within a time span of 10-15 minutes, too short to introduce profound changes in ON/OFF timing. 

Comment 2: Secondly, the information on medication and the assessment of the extent of the scurvy is also vague; as you measure TUG, you should have indicated the extent of the scurvy, not whether it was present or not.

Response: We performed the TUG according to the standard TUG test procedure as specified in Podsiadlo et al. 1991 (Reference 3 in our manuscript). We did not systematically assess the extent of scurvy and we also don't understand why a potential comorbid vitamin C deficiency would invalidate our results. Zooming in on scurvy is really irrelevant to the main objectives of our study, like it would be for many similar published studies validating TUG or other clinical tests. Regarding information on medication, please see our previous response.  

Comment 3: Finally, the TUG is not oriented. This leads to a lack of adjustment for confounding factors, which would make it difficult to state the results clearly.

Response: Indeed, we did not dictate the direction of the turns in the TUG, thereby adhering to the standard TUG test procedure as detailed in Podsiadlo et al. 1991. We did -based on the reviewer's remark- look for the presence of a confounding effect in that regard. For turn 1 of the TUG, the turn direction (i.e., clockwise or counterclockwise) was the same for test and retest in all our participants. The absence of a standardized instruction regarding turning direction did thus NOT introduce a confounding factor for turn 1. For turn 2 of the TUG (the one that is often combined with the sitting down manoeuvre), the turn direction was the same for test and retest in 18 of our 21 participants. Excluding these three participants from the analysis only marginally improved the test-retest reliability results (see Table below), so also here the lack of standardizing the turn direction did not introduce a confounding factor influencing the results.

We have added the following passage to the Results section 3.2 to describe the high consistency in turn directions between test and retest across our participants and how minor deviations in this for turn 2 only marginally affected the test-retest reliability statistics, thereby more clearly stating our results: “Note that we used the standard TUG, thus without dictating turning directions (i.e., clockwise or counter-clockwise). Nevertheless, turning directions of test and retest were highly consistent among our participants. For turn 1 at or around the 3-meter marker, all our participants turned in the same direction for test and retest. For turn 2 prior to sit-ting down, 18 of the 21 participants turned in the same direction for test and retest; excluding the 3 participants with an inconsistent turn direction slightly improved test-retest reliability statistics for turn 2 sub-durations (limits of agreement: from -0.14 (-1.28 1.00) to -0.02 (-1.06 1.02) and ICC(A,1) from 0.684 to 0.742).” 

Reviewer 2 Report

Comments and Suggestions for Authors

This manuscript has been sufficiently revised and has addressed all of the revisions requested. 

Thank you to the authors for addressing their reason for using two AR headsets and completing the follow-up analysis between AR systems. The between system agreement is encouraging and offers an initial validation into the accuracy of the Magic Leap 2. Inclusion of the vSLAM algorithm explanation and citation also offers readers information necessary to guide decisions when selecting an augmented reality headset to use for clinical or research use.

Author Response

Thank you for your positive review of our manuscript. We are pleased to hear that the revisions have satisfactorily addressed all the requested changes. Your feedback has helped to enhance the quality and clarity of our manuscript.